# High-Index Epitaxial Fe Films Grown on MgO(113)

**DOI:** 10.3390/ma16124352

**Published:** 2023-06-13

**Authors:** Wenzhi Peng, Yulong Chen, Xuhao Yu, Dazhi Hou

**Affiliations:** 1Hefei National Research Center for Physical Sciences at the Microscale, University of Science and Technology of China, Hefei 230026, China; pwz@mail.ustc.edu.cn (W.P.); cyul@mail.ustc.edu.cn (Y.C.); xhyu@mail.ustc.edu.cn (X.Y.); 2Department of Physics, University of Science and Technology of China, Hefei 230026, China

**Keywords:** crystal orientation, high-index, Fe(103), MgO(113)

## Abstract

The epitaxial growth of high-index Fe films on MgO(113) substrates is successfully achieved using direct current (DC) magnetron sputtering, despite the significant lattice constant mismatch between Fe and MgO. X-ray diffraction (XRD) analysis is employed to characterize the crystal structure of Fe films, revealing an Fe(103) out-of-plane orientation. Furthermore, our investigation reveals that the Fe[010] direction is parallel to the MgO[$1\bar{1}0$] direction within the films plane. These findings provide valuable insights into the growth of high-index epitaxial films on substrates with large lattice constant mismatch, thereby contributing to the advancement of research in this field.

## 1. Introduction

In materials science, a crystal facet refers to a flat surface that is bounded by crystallographic planes with specific Miller indices. High-index facets are those with indices greater than the standard low-index facets of a crystal, which are typically represented by (100), (110), and (111) in cubic crystals, for example. High-index facets are, in principle, infinite and can offer richer surface structures and properties than low-index facets. However, their controlled preparation is challenging, as they are neither thermodynamically nor kinetically favorable compared to low-index facets [1]. The pursuit of producing single-crystal films with various facet indices has been a long-standing goal in materials science, given their potential applications in crystal epitaxy, catalysis, electronics, and thermal engineering [2,3,4]. Recent advancements in electrochemical reactions, electrodeposition, and solution-phase synthesis have made it possible to obtain high-index metal foils and nanocrystals by perturbing the thermodynamic equilibrium state or modifying the kinetic barrier that hinders growth [5,6]. So far, there are few studies on how to grow high-index ferromagnetic thin films.

Epitaxial growth of Fe on MgO is a well-established technique used to produce high-quality Fe single crystals with an out-of-plane orientation of [001] and an in-plane orientation of Fe[1$\bar{1}$0] parallel to MgO[100] [7,8]. This technique has significant implications in the context of the Fe/MgO/Fe magnetic tunnel junctions (MTJs) structure due to its potential for enabling high-density data storage, low power consumption, and compatibility with established fabrication methodologies [9,10,11]. Moreover, the unique spin-dependent transport properties exhibited by MgO-based MTJs have resulted in their remarkable giant magnetoresistance effect, surpassing 400%, which is of great significance for its research both in theory and practical applications [12,13,14]. Although extensive research has been conducted on Fe/MgO(001)/Fe MTJs, there is still a lack of studies on high-index Fe/MgO/Fe MTJs. Exploring the feasibility and properties of high-index Fe/MgO/Fe MTJs represents an intriguing direction for future investigations. Despite the large difference in lattice constants between Fe and MgO, it is still possible for Fe(001) to achieve epitaxial growth on MgO(001) by matching the surface structure of Fe(001) with the crystal plane of MgO(001) when rotating its crystal orientation by 45°. Such a 45° rotation scheme cannot be easily adopted by other facets of Fe to realize epitaxial growth on MgO, especially when MgO is of a high-index orientation. It is still an open question whether single-crystalline Fe can be grown on high-index MgO.

In this study, we investigate the growth of high-index single crystal Fe films on MgO(113) substrates using DC magnetron sputtering. Our goal is to explore the feasibility of achieving epitaxial growth of Fe films on a high-index substrate despite the significant lattice constant mismatch between Fe and MgO. We employ XRD analysis to characterize the crystal structure of the Fe films and determine their orientation relationship with the substrate. The XRD results reveal that the Fe films exhibit a single-crystalline structure with an out-of-plane orientation along the [103] direction. Additionally, within the films plane, we find that the Fe[010] direction is parallel to MgO[$1\bar{1}0$]. By performing reciprocal space mapping (RSM) analysis, we observe a small deviation angle of approximately 1.2° between the Fe(103) plane and the MgO(113) plane, indicating a deviation from strict epitaxial growth. To investigate the magnetic properties, we conduct magnetization measurements using a Superconducting Quantum Interference Device (SQUID). The SQUID measurements demonstrate that the magnetic easy axis of the Fe(103) films lies in the in-plane [010] direction. Our study provides valuable insights into the growth behavior of Fe films on MgO substrates with high-index orientations, highlighting the potential for achieving epitaxial growth of high-index films in systems with significant lattice mismatch.

## 2. Materials and Methods

### 2.1. Materials and Reagents

Fe target with a purity of 99.99% was obtained from Hefei Kejing Company (Hefei, China), a renowned supplier in the field. The MgO(113) substrate, processed from MgO bulk material with a (110) crystallographic orientation and a roughness within the range of 0.5 nm (measured within a 5 × 5 μm^2^ area), was also sourced from Hefei Kejing Company.

### 2.2. Fe/MgO(113) Thin Films Fabrication

Fe/MgO(113) thin films were fabricated via DC magnetron sputtering. A high-purity Fe target (99.99%) was sputtered with Ar gas at a pressure of 5 × 10^−3^ mbar while maintaining a background vacuum of 1 × 10^−7^ mbar. Prior to deposition, the substrate was annealed at 600 °C for 1 h. The Fe layer was grown at 300 °C and annealed for 1 h at the same temperature after the growth process [8]. The sputtering power was set to 50 W and the growth rate was 1.09 Å/s. During the films growth process, the sample holder carrying the substrate was rotated continuously.

### 2.3. Fe/MgO(113) Thin Films Characterization

The Fe/MgO(113) thin films were subjected to comprehensive characterization using XRD and SQUID. For XRD analysis, the thin films’ in-plane and out-of-plane crystal orientations were determined. XRD measurements were conducted with a Cu target X-ray source, utilizing a minimum beam size of 0.4 × 8 mm^2^. The scanning range of 2*θ* spanned from 0° to 150°, while the adjustment range of χ covered −3° to 90°. Additionally, the adjustment range of φ extended from −180° to 180°. SQUID was employed to investigate the magnetic properties with exceptional sensitivity, reaching up to 10^−9^ emu. The measurements were performed under a maximum applied magnetic field of 7 Tesla, while maintaining a high magnetic field homogeneity of 0.01% within a 4 cm range. By utilizing XRD and SQUID, the crystal orientations and magnetic properties of the Fe/MgO(113) thin films were accurately characterized.

## 3. Results and Discussion

Figure 1a depicts the XRD spectra obtained from a 400 nm-thick Fe films grown on the MgO(113) substrate. The sample was leveled based on the maximum value of the MgO(113) diffraction peak. The obtained spectra shows a (103) peak of the Fe films, which exhibits a favorable azimuthal relationship to MgO(113), indicating good single crystalline quality. Figure 1b displays a *χ* scan in the vicinity of the Fe(103) peak, which reveals that the peak maximum for Fe(103) differs by approximately 1.2° from that of MgO(113). This observation suggests that the angle between the orientations of Fe(103) and MgO(113) is around 1.2°. Furthermore, Figure 1c shows the rocking curves of the (103) peak of the Fe films and the (113) peak of the MgO substrate, each having a full width at half-maximum (FWHM) of 0.24° and 0.02°, respectively. The comparison of their FWHM values implies that the Fe(103) thin films has excellent epitaxial quality.

To determine the in-plane crystal orientation, we carefully controlled the experimental conditions by setting the values of 2*θ*, *ω* and *χ* to 99.0°, 49.5° and 76.17°, respectively [15]. By rotating *φ* in the (xy) plane, we obtained Figure 1d. Notably, we observed that the peaks (2$\bar{2}$0) and (220) were located at 46.9° and −46.9°, respectively, which indicates that they exhibit mirror symmetry relative to the (xz) plane of the sample. To further illustrate our findings, we plotted Figure 1e to depict the directions of the observed (103), (220), and (2$\bar{2}$0). Furthermore, we inferred that the direction of <y> is [010] using vector subtraction. Additionally, based on orthogonality, we inferred that the direction of <x> is [30$\bar{1}$]. These essential observations convincingly determine the epitaxial relation between the Fe films and MgO(113).

Figure 2a illustrates the atomic structure of the (001) plane of Fe in a perfect crystal. The Fe atoms at the interface with MgO(001) are marked in purple. In Figure 2b,c, the atomic structure of the (103) plane of Fe and the (113) plane of MgO in a perfect crystal is depicted. The Fe, Mg, and O atoms at the Fe/MgO interface are denoted by black, blue, and red colors, respectively. The atoms in the adjacent layers at the Fe/MgO interface are shown in lighter shades to differentiate them. The Fe(103) crystal has a closest atomic distance of 2.867 Å, matching the lattice constant of BCC (Body-Centered Cubic) Fe, and a second closest atomic distance of 4.752 Å. The angle between these two directions is approximately 72.45°. In contrast, the MgO(113) crystal has a closest atomic distance of 2.978 Å and a second closest atomic distance of 5.159 Å, with an angle of approximately 73.22° between the two directions. This indicates that there is a lattice mismatch between Fe(103) and MgO(113) of approximately 7.9% and 3.7% in the closest and second closest directions, respectively. Despite the lattice mismatch between Fe(103) and MgO(113), our results shows the feasibility of growing high-quality single-crystalline Fe successfully on the surface of MgO(113). Within the high-index Fe(103) plane, there is only one mirror plane, sharing the same symmetry with MgO(113). It is different from the common Fe(001) system that has four mirror planes, as depicted in Figure 2a–c.

Figure 3 presents a detailed investigation of the strain states of Fe films using RSM techniques around the (103) diffraction. The *Q_x_* of Fe(103) and MgO(113) in Figure 3a,c are very close, while in Figure 3b,d they show finite deviation, indicating that Fe(103) and MgO(113) do not grow coherently. The results reveal that the lattice constant of the Fe thin films grown on MgO (113) is 2.867 Å, while the lattice constant of MgO (113) is 4.212 Å. Since the lattice constant of bcc Fe single crystalline is also 2.867 Å and that of MgO single crystalline is 4.212 Å, the Fe films is not strained much [16,17,18]. The data presented in Figure 3b,d suggest that there is a non-zero angle between the Fe(103) and MgO(113) planes, which is consistent with the results of Figure 1b.

Figure 4a shows the hysteresis loops of 100 nm-thick Fe(001) thin films measured by SQUID. The saturation fields for the [001], [$\bar{1}1$0], and [110] directions are 40,000 Oe, 40 Oe, and 40 Oe, respectively [19]. Figure 4b shows the hysteresis loops of 400 nm-thick Fe(103) thin films. The saturation fields for the [103], [$\bar{3}$01], and [010] directions are 35,000 Oe, 95 Oe, and 40 Oe, respectively. These *M-H* curves indicate that the out-of-plane direction of Fe films represents a hard axis due to shape anisotropy, and there is a finite uniaxial magnetic anisotropy with the Fe(103) plane. Due to the negligible in-plane shape anisotropy, the magnetic crystal anisotropy dominates, resulting in the [010] direction being the in-plane easy axis, and the [$\bar{3}$01] direction being the in-plane hard axis. Consequently, there is almost no residual magnetism in the [$\bar{3}$01] direction under zero-field conditions in Fe(103) plane.

To further manipulate the in-plane magnetic anisotropy of the Fe(103) films, we halted the rotation of the sample holder which induced oblique-angle deposition during the sputtering process. As a result, we obtained a 100 nm-thick Fe(103) films; the *M-H* curve is shown in Figure 4c. Additional measurements of the magnetic hysteresis loops in different directions revealed that the saturation field in both the in-plane [010] and [$\bar{3}$01] directions was approximately 600 Oe, with residual magnetism present at zero field, as shown in Figure 4c. Interestingly, while the in-plane saturated field increased, the out-of-plane saturated field decreased to 30,000 Oe. This controlled magnetic anisotropy might be attributed to the formation of columnar crystals within the Fe films induced by oblique incidence [20,21,22].

In oblique angle deposition, magnetic thin films typically exhibit inclined columnar structures due to the ballistic shadowing effect. As a result, the growth direction of the inclined columnar crystals deviates from being perpendicular to the substrate, leading to an easy plane that is offset from the substrate at a certain angle. The direction along which the particles are deposited in the thin-films plane is known as the hard axis, while the direction perpendicular to the deposition direction is referred to as the easy axis [23,24].

In the case of Fe(103) thin films, which possess single-axis anisotropy within the Fe(103) plane, the oblique angle deposition along the Fe[010] direction can weaken or even cancel out the single-axis anisotropy within the Fe(103) plane. This phenomenon occurs because the inclined columnar growth introduces a tilt in the crystallographic alignment, altering the magnetic properties of the films. Consequently, the saturation field along both the Fe[010] and Fe[$\bar{3}$01] directions, as depicted in Figure 4a, is approximately 600 Oe. The observed offset easy plane and the reduction or elimination of single-axis anisotropy in obliquely deposited Fe(103) films have significant implications.

## 4. Discussion

To provide clearer and more effective guidance for future experiments, we have considered different substrate orientations and surface structures in our study. In addition to the Fe(103)/MgO(113) system discussed in this paper, we explored the theoretical possibility of growing Fe single crystals on MgO(203) substrates with an outward orientation of Fe(223). As depicted in Figure 5a,b, the surface structures of these two interfaces exhibited remarkable similarity. The lattice mismatch between Fe(223) and MgO(203) along the nearest neighbor and second nearest neighbor directions is approximately 9.1% and 3.8%, respectively.

Drawing a comparison between the surface structures of substrates and thin films with different crystallographic indices provides a convenient and practical approach to preliminarily determine the orientation of single-crystal thin films grown on various substrate crystal planes, especially when there are significant differences in lattice constants between the substrate and the thin films. This method allows for a straightforward assessment of the potential orientation of thin-films single crystals grown on different high-index surface planes.

Moreover, considering the use of substrate materials that offer better lattice constant matching, such as Pt, on high-index MgO surfaces, greatly facilitates the prediction of single-crystal orientations. The relatively small difference in lattice constants between MgO and Pt enables the growth of Pt(001) on MgO(001) and Pt(111) on MgO(111), achieving strict epitaxial growth. Due to the similarities in lattice constants and crystal symmetries between MgO and Pt, it is reasonable to anticipate the growth of Pt(113) on MgO(113) as well. Despite the extensive research on the Pt(001)/MgO(001) structure, there remains a lack of effective detection of spin transport measurements on high-index Pt/MgO interfaces. Therefore, investigating this direction represents a valuable opportunity for further exploration.

By exploring alternative substrate orientations and considering materials with improved lattice constant matches, we can gain a deeper understanding of crystal growth mechanisms and optimize the design of thin-films structures for specific applications. Furthermore, the investigation of novel interfaces, such as Pt/MgO on high-index surfaces, not only expands our fundamental knowledge but also holds great potential for advanced device fabrication and spintronic applications.

In the case of XRD characterization of thin films under special circumstances, certain crystallographic planes may not be observable due to diffraction interference, posing a complex and intriguing challenge. For example, in the BCC crystal structure, when the sum of the Miller indices (hkl) is an odd number, destructive interference occurs and the corresponding XRD peaks cannot be detected. In such situations, one possible approach is to employ tilted XRD measurements. Although conventional XRD measurements cannot provide information about the crystallographic planes, tilting the sample or the detector allows changes to the incident and diffracted angles, thereby detecting XRD signals from other crystallographic planes of the thin films. By reconstructing the sample rotation angles, it becomes possible to infer the original orientation of the thin-films surface.

Materials with low symmetry often exhibit rich and intriguing physical properties due to the mirror symmetry breaking in their crystal structures. For instance, the In-plane Anomalous hall effect observed in heterodimensional superlattice and the out-of-plane anti-damping-like torques measured in MnPb_3_(114) highlight such phenomena [25,26]. However, a major drawback of these low-symmetry materials is the challenge in large-scale production with specific crystal orientations, hindering reproducibility for others. To address this issue, we propose a novel approach in this article: utilizing block material cutting to obtain high-index substrates, followed by epitaxial growth of single-crystal films. This method enables the fabrication of uniform, highly oriented thin films with unique physical properties owing to their inherently low symmetry. Thus, it encourages further extensive research in this exciting field.

## 5. Conclusions

Our study has successfully achieved the epitaxial growth of Fe thin films on MgO (113) substrate, as confirmed by our XRD analysis. Specifically, we determined that the epitaxial relationship between Fe and MgO is Fe [103]∥MgO[113] and Fe[010]∥MgO[33$\bar{2}$]. Additionally, our analysis demonstrated that the (103) plane of Fe thin films tilts 1.2° angle from the MgO(113) plane, and we observed little lattice distortion in the Fe(103) films, with a lattice constant of 2.867 Å. Furthermore, we conducted a SQUID measurement to characterize the magnetic properties of the Fe thin films. High-index ferromagnetic thin films that have been successfully grown have potential value for subsequent transport, thermal, and epitaxial crystal applications.

To further facilitate future experiments, we predicted the growth of Fe (223) single-crystal thin films on MgO (203) substrates. This method is applicable for growing single crystal films on materials with mismatched lattice constants. By comparing the atomic structures of the substrate and the films with different crystallographic orientations, we can determine the feasibility of growing single-crystal thin films and the orientation of the film’s surface. For materials with matched lattice constants, such as MgO and Pt, the difficulty of growing high-index single crystal thin films is reduced, and the orientation of the films should align with that of the substrate. This suggests the possibility of growing Pt (113) on MgO (113) for spin transport measurements, providing a convenient approach to grow low-symmetry materials for spin transport studies.

## Figures and Tables

**Figure 1 materials-16-04352-f001:**
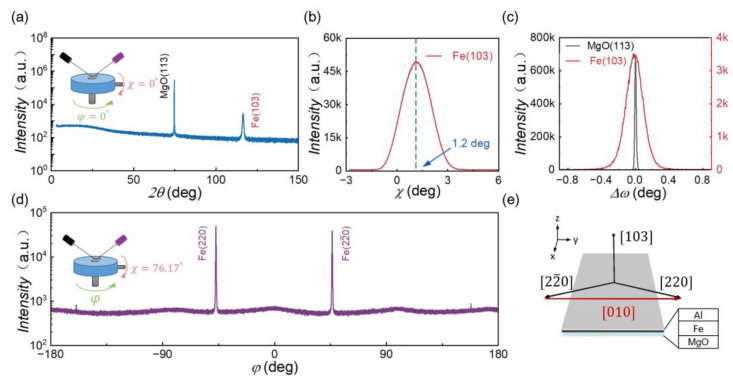
(**a**) XRD *2θ–ω* linear scan, (**b**) *χ* scan near the Fe(103) peak, (**c**) rocking curves, (**d**) and pole figure XRD spectra of Fe(220) and Fe(2$\bar{2}$0) from Fe(103) on MgO(113). (**e**) Crystal orientations of Fe(103)/MgO(113) in different orientations (black arrows), which can be utilized to obtain the in-plane crystal orientation (red arrows).

**Figure 2 materials-16-04352-f002:**
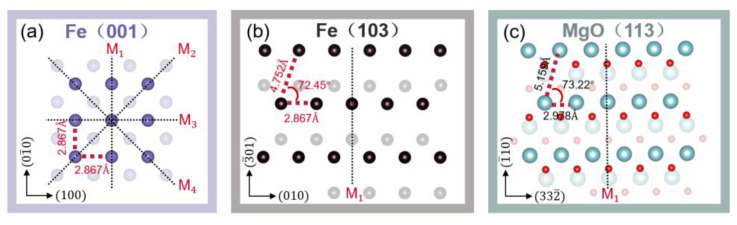
Schematic of (**a**) Fe(001), (**b**) Fe(103), and (**c**) MgO(113) surfaces.

**Figure 3 materials-16-04352-f003:**
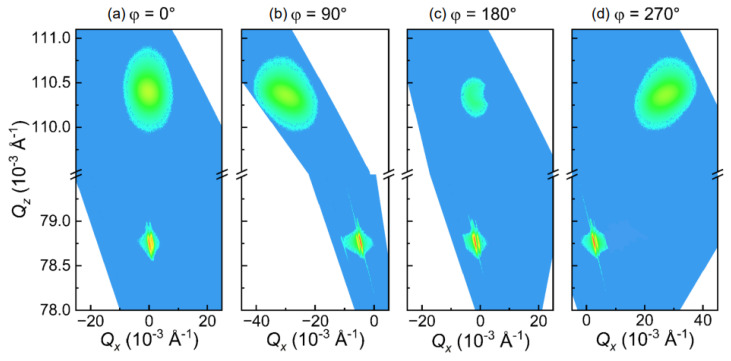
RSMs at different φ near the Fe(103) and MgO(113) peaks, with high-intensity diffraction peaks shown in red and low-intensity diffraction peaks shown in blue.

**Figure 4 materials-16-04352-f004:**
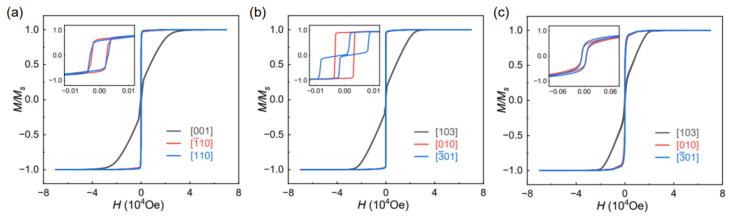
Magnetic hysteresis loops of Fe/MgO structures with different orientations. (**a**) 100 nm-thick Fe(001) on MgO(001). (**b**) 400 nm-thick Fe(103) on MgO(113). (**c**) 100 nm-thick Fe(103) on MgO(113) grown by oblique angle deposition. Insets are the low field results.

**Figure 5 materials-16-04352-f005:**
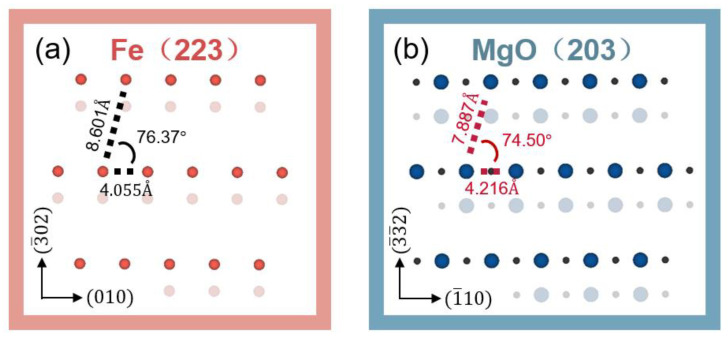
Schematic of (**a**) Fe(223) and (**b**) MgO(203) surfaces.

## Data Availability

Not applicable.

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
