# Peer review of "High-Index Epitaxial Fe Films Grown on MgO(113)"

_materials, 2023, doi:10.3390/ma16124352_

Round 1

Reviewer 1 Report

I have now read the manuscript titled: “High-index epitaxial Fe film grown on MgO” submitted to the Journal of “Materials”. The authors investigated the epitaxial growth of Fe thin film on MgO substrate, i.e., showing the relationship between Fe and MgO. They used the XRD technique to characterize and confirm the crystal structure of the end product. Detailed comments are below.

 - Please improve the language of the manuscript.

- The abstract part needs to be improved and data support.

- The introduction part needs to be revised as it is too short. At the end of the Introduction section, there is no clear explanation of the scientific contribution of this paper. What is the innovation of the article? How is it different from previous research? I think that the introduction is missing the goals of the paper. Hence, I suggest revising the last paragraph of the introduction and clearly stating the goals and novelty of the presented research.

- Please add more information about the details of the experiments.

-If possible, please add the images of the Fe/MgO thin film samples in the manuscript.

-Please add quantitative results in the conclusion part and improve it.

-My final comment is about the references part. I think this part needs to be improved and add more recent references.

- Please improve the language of the manuscript.

Author Response

Our response can be found in the attachment. Thank you.

Reviewer 2 Report

This manuscript presents the epitaxial growth of high-index single crystalline Fe film on MgO(113) substrates. The authors have extensively characterized the crystal orientation of the Fe film using XRD, and analyzed the strain states during the film growth. Additionally, the magnetic properties of the Fe films have been studied and can be manipulated by altering the growth conditions. Overall, this comprehensive study can serve as a useful reference for future research on epitaxial growth of high-index films on MgO. However, there are some typos in the manuscript, and I recommend the authors to carefully review the grammar throughout the text:

Typos include:

Line 37: Change “growth on MgO by matching of the surface structure” to “growth on MgO by matching the surface structure”

Line 50: change “an deviation angle” to “a deviation angle”

Line 125: change “unaxial” to “uniaxial”

Line 126: change “anistropy” to “anisotropy”

1. This research addressed the epitaxial growth of Fe on high-index MgO surface.

2. This topic is relevant to the field of epitaxial growth and it reports that Fe can be grown on high-index MgO surface. However, I don't think this work addresses the gap in this field by reporting a single example.

3. Most published materials report epitaxial growth on low-index MgO surface. It is intrinsically difficult to grow Fe on high-index MgO surface due to the structural mismatch while this manuscript has achieved this goal. However, the authors should provide some references regarding the film growth on high-index MgO surface.

4. I think the methodology is robust.

5. As mentioned in the third answer, the authors should provide some references regarding the film growth on high-index MgO surface.

6. The font size in Figure 1 is too small. I suggest increasing it.

Author Response

(The authors gave the same response as above.)

Reviewer 3 Report

The presented manuscript includes the study of the high-index epitaxial Fe film grown on MgO(113). 

The results of the work are presented on a good level, the text is well structured, and the figures are clear and well treated, but some questions and weak points regarding paper structure should be mentioned. The paper is of interest and fits the Materials journal.

1. Please, do not use more than 3 references in one place (like [2-6]). Either you should describe the differences. 

2. Please, define the abbreviation “DC” in the text before its first mention.

3.   In order to make your experiments reproducible by other researchers please split section 2. “Materials and methods” to the 3 separate subsections. First SubSection 2.1 as “Materials and reagents” (here please mention all materials and reagents used in your study, their purity, and supplier), second as 2.2 “Fe/MgO(113) thin films fabrication”, third as 2.3 “Fe/MgO(113) thin films characterization” (here please mention all methods and instruments as well as selected parameters for analysis).  

Author Response

(The authors gave the same response as above.)

Round 2

Reviewer 1 Report

I just read the manuscript (revised version) titled: “High-index epitaxial Fe film grown on MgO” submitted to the Journal of “Materials”. The authors investigated the epitaxial growth of Fe thin film on MgO substrate, i.e., showing the relationship between Fe and MgO. They used the XRD technique to characterize and confirm the crystal structure of the product. The authors improved the language of the revised manuscript. The new explanations about the materials part were added in the revised part. I think it can be accepted in this revised form.

Reviewer 2 Report

The authors have addressed my comments.

Reviewer 3 Report

all coments were addressed